# Antimicrobial Activity In Vitro of Cream from Plant Extracts and Nanosilver, and Clinical Research In Vivo on Veterinary Clinical Cases

Teodora P. Popova [1], Ignat Ignatov [2,*], Toshka E. Petrova [1], Mila D. Kaleva [3,4], Fabio Huether [5] and Stoil D. Karadzhov [6]

1 Faculty of Veterinary Medicine, University of Forestry, 10 Kl. Ohridski Boulevard, 1756 Sofia, Bulgaria
2 Scientific Research Center of Medical Biophysics (SRCMB), 1111 Sofia, Bulgaria
3 The Stephan Angeloff Institute of Microbiology, Bulgarian Academy of Sciences, 26 Akad. G. Bonchev Street, 1113 Sofia, Bulgaria
4 Veterinary Clinic "Union Vet", 5 Buenos Aires Street, 1172 Sofia, Bulgaria
5 EVODROP AG, 8048 Zurich, Switzerland
6 Bulgarian Association of Activated Waters, 1619 Sofia, Bulgaria
* Correspondence: mbioph@abv.bg

**Abstract:** The antimicrobial effect of a cream containing extracts of African geranium (*Pelargonium sidoides* DC.), black elderberry (*Sambucus nigra* L.), and St. John's wort (*Hypericum perforatum* L.) in colloidal nanosilver (AgNPs) at a concentration of 30 ppm, denoted as SILVER STOP® cream (SS® cream), was examined in vitro. The research was performed with *Escherichia coli* (ATCC and two clinical isolates), *Staphylococcus aureus* (ATCC and two clinical strains), and *Candida albicans* (ATCC and two clinical isolates). The agar-gel diffusion method and suspension tests for determination of the time of antimicrobial action of SS® cream were used. SS® cream showed significant antimicrobial activity. The Gram-negative microorganisms tested died in a much shorter time than the Gram-positive ones. In suspension with a density of $10^4$ cells·mL$^{-1}$, *E. coli* died for 1 min, the oval fungus *C. albicans*—after 10 min and *S. aureus*—after 60 min of exposure to SS® cream. The highest sensitivity was found in *E. coli*. The curative effect of SILVER STOP® cream was also examined in vivo in dogs with different skin diseases. The results showed successful healing of the diseases and a very good curative effect of the cream.

**Keywords:** colloidal nanosilver AgNPs; *Pelargonium sidoides*; *Hypericum perforatum*; *Sambucus nigra*; antimicrobial activity; dogs; skin diseases





## 1. Introduction

In recent decades, the interest of a number of research groups was focused on the study of alternative strategies for the development of new antimicrobial agents due to the ever-increasing microbial resistance against commonly used antibiotics [1–3]. A number of recent studies show that silver nanoparticles (AgNPs) exhibit antimicrobial activity and, thus, can be used to develop a new type of antimicrobial agent for the treatment of bacterial infections. Effects are also observed in bacterial infections caused by multidrug-resistant microorganisms [4–7]. On the other hand, nowadays, nanotechnology offers new approaches to the use of plants in order to create new bioactive drugs. Silver nanoparticles show therapeutic properties for the treatment of many diseases due to their antiplasmodial, antibacterial, and antifungal activity [4,8]. Results are achieved for SARS-CoV-2 as well [9]. Using plant extracts, it is possible to obtain safe, effective, cheap, and ecological drugs [10], and to improve the activity of dry plants [11], although these are used rarely as pharmaceutical or cosmetic raw materials. Silver nanoparticles are successfully applied as antibacterial agents and for the therapy of skin damage such as burns and ulcers [12]. Due to their small size, they have many beneficial properties such as high surface-to-volume ratio,

high chemical activity, significant antimicrobial and fungicidal activity, and biocompatible surface properties. These properties are attributed to the small size of nanoparticles, which is why the prospects for their diverse application nowadays are significant. The pharmaceutical industry uses them in antimicrobial and antifungal mixtures. Silver was used in ancient times to treat burns, wounds, and bacterial infections but today, it is being relied upon again due to the increasing antibiotic resistance of bacteria. Silver is also found to be non-toxic to humans at minimal concentrations [13]. The estimated level of consumption of silver in humans is around 20–80 µg per day [14,15]. For skin application, the colloidal nanosilver dosage is 0.01 mg·m$^{-2}$ [16]. The dose to be used for SILVER STOP$^®$ cream is 0.0092 mg·m$^{-2}$ [16]. This is the amount of silver in a one-time application. The resistance developed by microorganisms to silver is less, compared to that of antibiotics [17–19].

A number of plants exhibit antimicrobial properties in the context of the increasing resistance of microorganisms to antibiotics. The application of plant extracts is more and more taken into account for the treatment of various infections. A very good track record is achieved by African geranium (*Pelargonium sidoides* DC.), black elderberry (*Sambucus nigra* L.), St. John's Wort (*Hypericum perforatum* L.), etc. [17,20–22]. The health properties of elderberry fruits are known, which is why they are widely used in herbal medicine and the food industry. The antibacterial activity is proved of elderberry extracts against *Micrococcus luteus*, *Escherichia coli*, *Proteus mirabilis* и *Pseudomonas fragii*, which is due to the content of phenolic acids and flavonoids in them. *Pelargonium sidoides* exhibits moderate direct anti-infective properties but quite strong immunomodulatory activity [23]. Possibilities are shown for the successful application of *Hypericum perforatum* products in dermatology [24,25]. Oils or tinctures of St. John's Wort are applied topically to treat minor wounds and burns, sunburns, scrapes, bruises, cuts, ulcers, myalgia, and many others. The hypericin and hyperforin contained in it have antimicrobial, antioxidant and anti-inflammatory effects [26]. In addition, hyperforin stimulates the growth and differentiation of keratinocytes; however, clinical research in this area is still scarce. St. John's Wort has potential for use in medicinal skin care [25]. It is an established synergy of hyperforins and hypericins, as well as other ingredients such as flavonoids and biflavones, which may explain why traditional preparations seem to have a stable effectiveness despite their variable composition. The laboratory and clinical studies conducted to evaluate the antimicrobial and therapeutic properties for skin and other diseases of African Geranium, Black Elderberry, St. John's Wort, however, are insufficient. There are reports of successful traditional administration of 315 home-made herbal medicines used in cattle for treatment of skin diseases, wounds, etc. [26].

The aim of the present work was to perform studies to evaluate the antimicrobial activity of SILVER STOP$^®$ cream containing extracts of African geranium (*P. sidoides*), black elderberry (*S. nigra*), and St. John's Wort (*H. perforatum*) in colloidal nanosilver (AgNPs) at a concentration of 30 ppm against Gram-negative and Gram-positive microorganisms, some of the most common causes of difficult-to-treat infections in humans and animals. The second goal of the study was to show the usefulness of the preparation through its application in animals with skin infections.

## 2. Materials and Methods

*Antimicrobial product.* The antimicrobial effect of SILVER STOP$^®$ cream (SS$^®$ cream), prepared by the patent of Baiti [27], containing extracts from African geranium (*Pelargonium sidoides* DC.), black elderberry (*Sambucus nigra* L.), and St. John's Wort (*Hypericum perforatum* L.) [28] in colloidal nanosilver (AgNPs) [29] at a concentration of 30 ppm (from EVODROP AG, Zurich, Switzerland) was tested.

*Controls.* The broad-spectrum antibiotic thiamphenicol (Nikovet–Sofia) was used as a positive control, to which the tested microorganisms did not show resistance.

*Microorganisms.* Pure cultures of nine pathogenic strains were tested: two clinical strains of *Escherichia coli* and a reference *E. coli* ATCC-8739, two clinical strains of *Staphylococcus aureus* and the control *S. aureus* subsp. *aureus* ATCC-6538, as well as two clinical strains of *Candida albicans* and a reference *C. albicans* ATCC 10231 (NBIMCC 74). The clinical

strains had been isolated from cutaneous inflammatory secretions of dogs in the Laboratory of Microbiology at the University Clinic at the University of Forestry, Faculty of Veterinary Medicine in Sofia.

***Nutrient media.*** Mueller-Hinton agar and broth (BUL BIO NCIPD—Sofia, Bulgaria) and Columbia blood agar (Biolab Zrt. H-1141, Budapest Ov. Utra 43) were used, as well as selective media: Endo agar for *E. coli*, Chapman Stone agar for staphylococci and Sabouraud dextrose agar with chloramphenicol for *C. albicans* (Antisel—Sharlau Chemie SA, Barcelona, Spain), and Colorex Chromogenic Orientation agar (HiMeida Laboratories Pvt. Ltd., Mumbai, India).

The cultivation of the microorganisms was carried out at 35–37 °C for 18–24 h for the bacterial strains and 48–72 h for *C. albicans* strains under aerobic conditions.

***The antibacterial effect*** was studied using the classical agar-diffusion method of Bauer and Kirby [30], designed for fast-growing microbes under aerobic conditions, on Mueller-Hinton agar with pH = 7.2–7.4 and layer thickness 4 mm. SS® cream was applied by dropping of 0.1 mL in wells with a diameter of 9 mm. The control antibiotic Thiamphenicol was tested at a standard final concentration of 30 µg per well. Inoculation of the microbial suspensions at a dose of $2 \times 10^6$ cells·mL$^{-1}$ was followed by culturing at 35–37 °C for 18–24 and 72 h. The results were read by measuring the diameters of the inhibition zones in millimeters to the nearest whole millimeter, including the diameter of the well, with a transparent ruler on the outside of the bottom of the plates. As a boundary of the zone, complete suppression of growth was reported, without taking into account small colonies or barely visible growth, detected by close observation under a magnifying glass. The measured zones of inhibition were interpreted according to the three-stage categorization system of Bauer and Kirby. The susceptibility of the tested microorganisms to the studied preparation SS® cream was determined for non-antibiotic agents, such as sulfonamides, namely: resistant (R)—at areas with diameters < 12 mm, moderately or intermediate sensitive (I)—for zones within 13–16 mm, and sensitive (S)—at > 17 mm. For thiamphenicol the corresponding limits are: R < 12 mm, I—13–17 mm, and S—>18 mm [31,32].

*Determination of the Time of Antimicrobial Action of SS® Cream and Colloidal Nanosilver*

- A suspension of each of the tested microbial strains with a concentration of $10^5$ cells·mL$^{-1}$ in an amount of 0.1 mL was added to 0.9 mL of SS® cream, as well as to 0.9 mL of sterile water as a control of the microbial growth, where the final concentrations became $10^4$ cells·mL$^{-1}$.
- A suspension of each of the tested microbial strains with a concentration of $10^7$ cells·mL$^{-1}$ in an amount of 0.1 mL was added to 0.9 mL of SS® cream, as well as to 0.9 mL of sterile water as a control of the microbial growth, where the final concentrations became $10^6$ cells·mL$^{-1}$.
- The following controls were applied—sterile distilled water (without SS® cream) with the same content of each of the studied microbial strains, as well as SS® cream without microorganisms.

After homogenization for 1 min on a Vortex apparatus (Heidolph—Labimex, Sofia, Bulgaria) and different time intervals for exposure of the microorganisms to SS® cream tested (1 min, 5 min, 10 min, 20 min, 40 min, 60 min, 2 h, and 24 h) cultures were made from each of the samples on Mueller-Hinton agar and on the corresponding selective nutrient media according to the types of the microorganisms tested, which were cultured at 37 °C for 24–48 h under aerobic conditions. After cultivation, the growth of the tested bacteria was counted and the number of colonies developed was determined.

All the experiments were performed three times.

***The statistical processing*** of the results was performed according to the classical method of Student–Fisher with *t*-test. Microsoft® Office Professional Plus Excel 2013 (15.0.4569.15060) was used for the calculations, with rights from the University of Forestry, Sofia. The average values and their standard deviations were calculated. Student's *t*-test analysis for independent samples was applied to determine the statistical dependence

and reliability of the results. The Student's *t*-test was counted for 3 results in each group. Significance of the differences was defined at significance level $p < 0.05$.

Clinical cases with skin pathologies treated with SS® cream.

*Patient 1.* A dog named Ivancho, cocker spaniel breed, 14 years old, male. It does not have any accompanying systemic diseases. During a routine clinical examination in the lumbar region, established is disseminated mycotic dermatitis, accompanied by several hairless patches with uneven edges, desquamated skin and mild erythema. Microbiological and microscopic examinations confirm the diagnosis of **microsporia** (*Microsporum* spp.).

*Patient 2.* A dog named Jimmy, cocker spaniel breed, 12 years old, male. It does not have any accompanying systemic diseases. Admitted to the clinic with bleeding from the perianal area. Examination reveals swelling and profuse bleeding from the fistula with irregular and imbibed edges. The area is repeatedly treated with Granofurin. The pathoanatomical diagnosis is **hepatocellural adenoma.**

*Patient 3.* A dog named Naya, female. No indications of any accompanying diseases. A several-years-old hairless spot in the tarsal area. Microbiological and microscopic examinations prove a diagnosis of **ringworm** (*Trichophyton* spp.).

*Patient 4.* Dog. Male. There are no data for accompanying diseases. On a routine examination, severe inflammation in the chin area was revealed. The clinical condition was accompanied by severe erythema, folliculitis, multiple crusts, and significant alopecia in the area (**Pyoderma superficialis**). By the microbiological examination, a mixed staphylococcal infection with a predominance of *Staphylococcus schleiferi* was shown.

## 3. Results

The summarized results reflecting the effect of SS® cream on the experimental strains, determined by the agar-gel diffusion method, are presented in Table 1. The data show that the growth of all strains was successfully suppressed by SS® cream and the control antibiotic. The diameters of the inhibitory zones when applying SS® cream varied from $15.3 \pm 0.4$ to $19.0 \pm 2.0$ mm. The tested strains of *E. coli* showed the highest sensitivity and the lowest one—strains of *C. albicans* and *S. aureus*, but the differences were not significant ($p > 0.05$). The susceptibility of the studied microorganisms to the control antibiotic was higher than that to SS® cream, but the differences were not significant ($p > 0.05$).

**Table 1.** In vitro antimicrobial activity of SS® cream on *Escherichia coli*, *Staphylococcus aureus*, and *Candida albicans*.

| Microorganisms | No of Strains | Inhibitory Zones in mm | |
|---|---|---|---|
| | | SS® Cream | Thiamphenicol |
| *E. coli* | 3 | $19.0 \pm 2.0$ | $26.3 \pm 4.9$ |
| *S. aureus* | 3 | $15.6 \pm 1.6$ | $25.0 \pm 1.4$ |
| *C. albicans* | 3 | $15.3 \pm 0.4$ | $23.7 \pm 4.0$ |

The results from the studies by the suspension method showed that the tested SS® cream exhibited significant antimicrobial activity. As can be seen from the data in Table 2, when in suspension with a density of $10^4$ cells·mL$^{-1}$, the studied Gram-negative bacteria (*E. coli*) died quickly—for 1 min. After 5 min of exposure to SS® cream, the amount of *S. aureus* decreased by about 44% compared to the untreated control, and after 40 min only single cells remained viable in some samples. No growth of *S. aureus* was observed after 60 min of exposure to SS® cream. In *C. albicans*, the effect of the cream was faster, and after 10 min, there was also no growth of the strains exposed to the drug.

**Table 2.** Antimicrobial effect of SS® cream on Gram-positive and Gram-negative microorganisms in suspensions with a density $10^4$ cells·mL$^{-1}$.

| Microorganisms | Growth of the Strains (Percent of Colony Number in Comparison with the Untreated Controls) after Different Intervals of Exposure | | | | | | |
|---|---|---|---|---|---|---|---|
| | 1 min | 5 min | 10 min | 20 min | 40 min | 60 min | 2 h |
| *Escherichia coli* | 0 | 0 | 0 | 0 | 0 | 0 | 0 |
| *Staphylococcus aureus* | 74.3 ± 4.2 | 56.0 ± 4.3 | 35.0 ± 4.1 | 19.7 ± 6.1 | 3.8 ± 0.1 | 0 | 0 |
| *Candida albicans* | 79.0 ± 2.9 | 49.3 ± 2.5 | 0 | 0 | 0 | 0 | 0 |
| *Untreated controls* | 100.0 | 100.0 | 100.0 | 100.0 | 100.0 | 100.0 | 100.0 |

When in suspension with a density of $10^6$ cells·mL$^{-1}$ (Table 3), under the action of SS® cream, the quantity of the studied Gram-negative bacteria (*E. coli*) was reduced by about 80% within 1 min. After 5 min, only single colonies were detected, and after 10 min of exposure to the medicine tested, all the cells were killed.

**Table 3.** Antimicrobial effect of SS® cream on Gram-positive and Gram-negative microorganisms in suspensions with a density $10^6$ cells·mL$^{-1}$.

| Microorganisms | Growth of the Strains (Percent of Colony Number in Comparison with the Untreated Controls) after Different Intervals of Exposure | | | | | | |
|---|---|---|---|---|---|---|---|
| | 1 min | 5 min | 10 min | 20 min | 40 min | 60 min | 120 min |
| *Escherichia coli* | 19.3 ± 6.7 | 6.0 ± 4.3 | 0 | 0 | 0 | 0 | 0 |
| *Staphylococcus aureus* | 88.0 ± 1.6 | 82.3 ± 2.1 | 72.3 ± 2.1 | 62.7 ± 3.8 | 53.3 ± 4.7 | 26.7 ± 12.5 | 7.7 ± 2.1 |
| *Candida albicans* | 85.7 ± 3.3 | 73.3 ± 2.5 | 60.3 ± 3.7 | 39.0 ± 2.9 | 20.7 ± 2.5 | 0 | 0 |
| *Untreated controls* | 100.0 | 100.0 | 100.0 | 100.0 | 100.0 | 100.0 | 100.0 |

*S. aureus* reacted slowly. After 50 min of exposure to SS® cream, the amount of *S. aureus* decreased by about 47% compared to the untreated control, and after 2 h, only single cells remained viable in some samples. No growth of *S. aureus* was observed after 24 h of exposure to SS® cream. In *C. albicans*, the effect of SS® cream was faster than that of *S. aureus*. The quantity of the cells was reduced by about 80% within 40 min, and after 1 h, no growth was found.

Clinical cases with skin pathologies treated with SS® cream:

*Patient 1.* Hair was removed from the affected area. SS® cream was prescribed for application twice a day until healing. The owners did not report any irritation caused by the cream. The treatment lasted for 30 days, and on the 20th day of the therapy, the patient comes for a control examination. There was no erythema and desquamated skin. A total of 49 days after the beginning of the therapy, complete restoration of the hair cover in the affected area was noted. Particularly striking was the darker pigmentation of the hair. The photos (Figure 1) show subcutaneous formations that are age related.

*Patient 2.* SS® cream was prescribed twice a day until recovery. A visible result was noticed already on the 7th day when the opening of the fistula and the swelling in the area were decreased. The application of the preparation lasted for 45 days. At the end of the period, the fistula was completely closed and the swelling disappeared (Figure 2). Owners reported no irritation caused by the cream.

20 May 2022

30 May 2022

18 July 2022

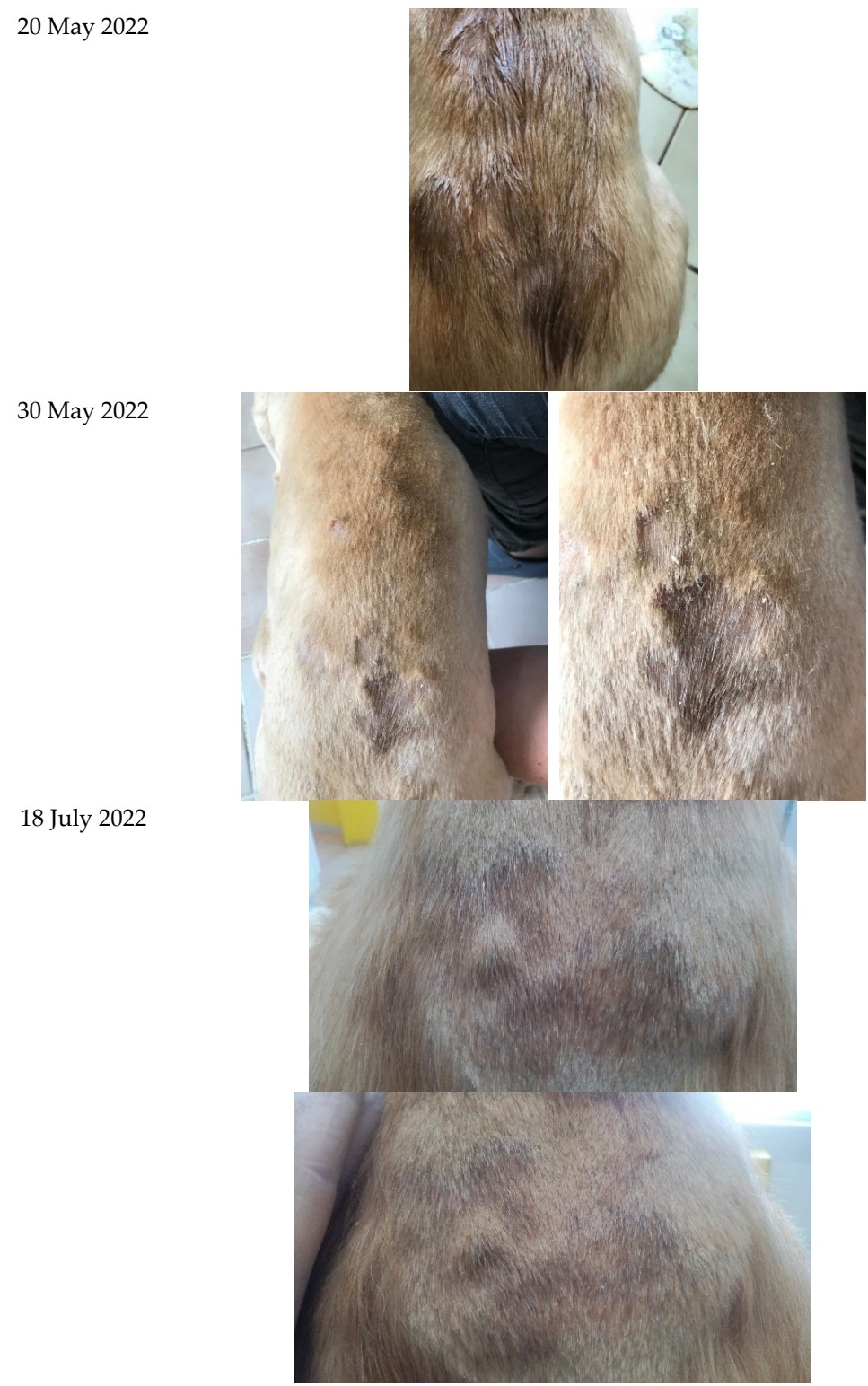

**Figure 1.** Treatment results of patient 1 with SILVER STOP® cream.

31 May 2022

01 June 2022

08 June 2022

16 July 2022

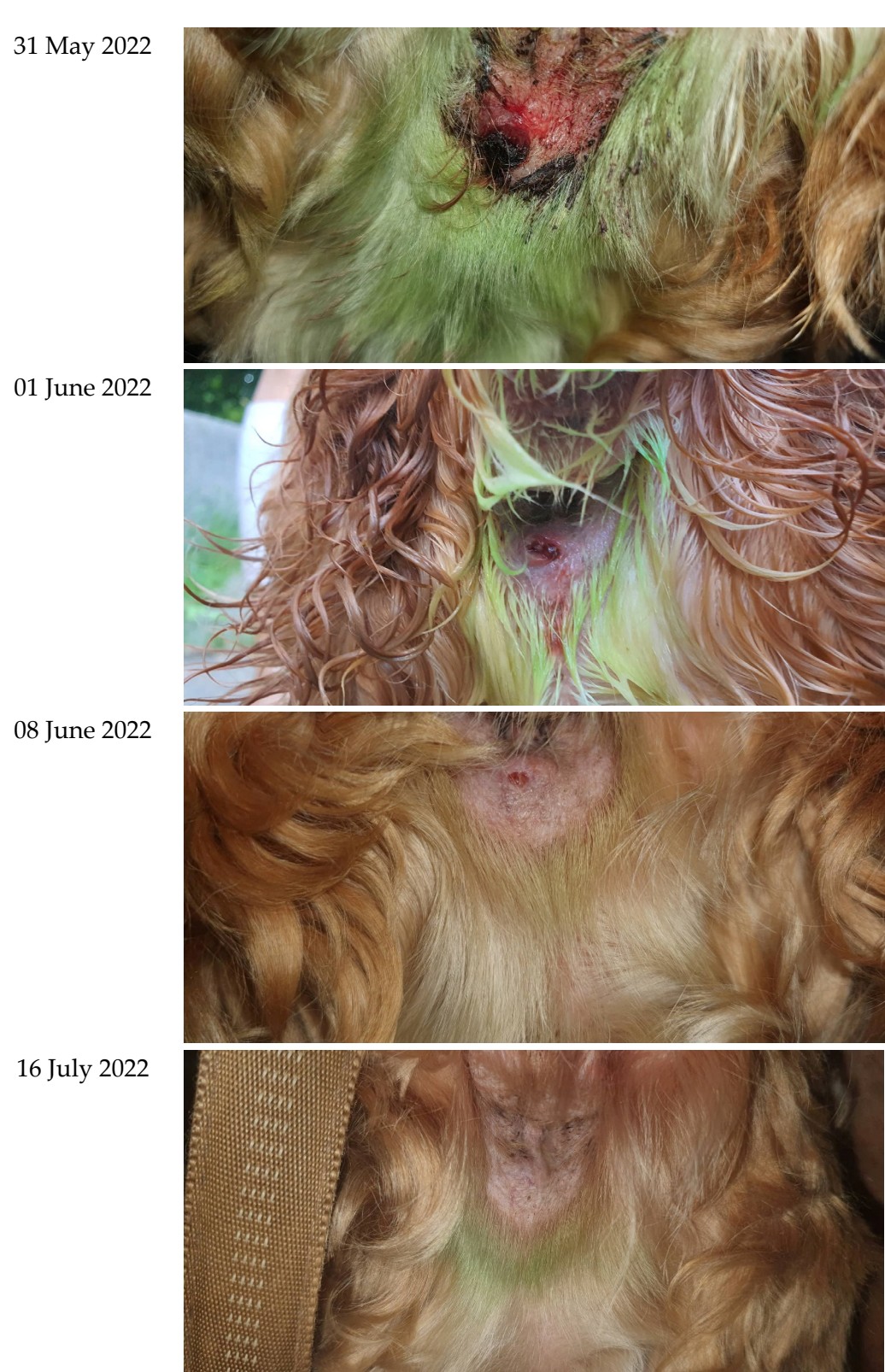

**Figure 2.** Treatment results of patient 2 with SILVER STOP® cream.

*Patient 3.* SS® cream was applied twice a day for 10 days. At the end of the therapy, a decrease in the diameter and a darkening in the pigmentation of the skin in the treated area (Figure 3) was observed. The condition started 15 months ago without any signs of improvement so far. Owners report no irritation caused by the cream.

3 April 2022

30 April 2022

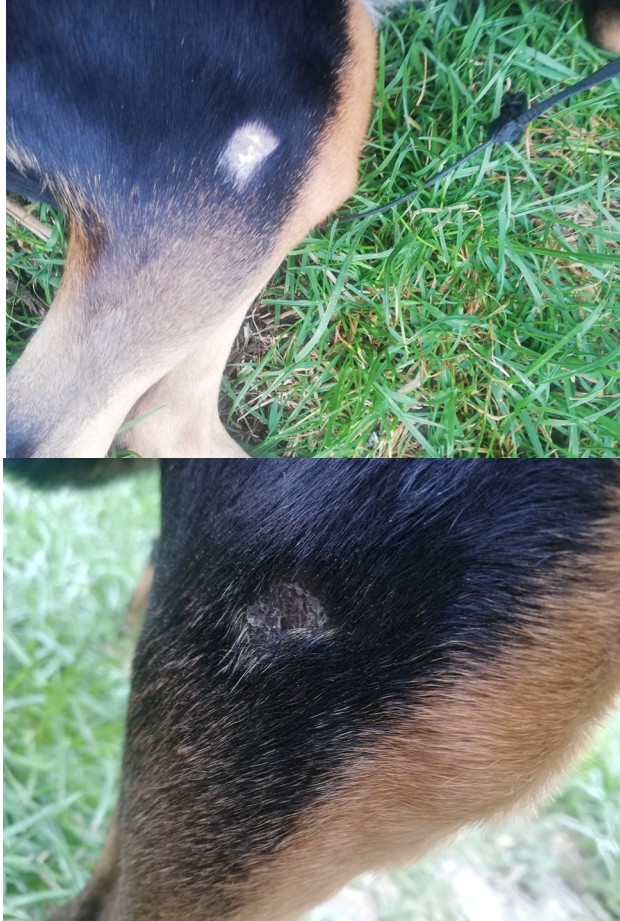

**Figure 3.** Treatment results of patient 3 with SILVER STOP® cream.

*Patient 4.* SS® cream was prescribed for application twice daily for 15 days. At the end of the period, there was an absence of crusts, folliculitis, and erythema. New hair growth was observed (Figure 4).

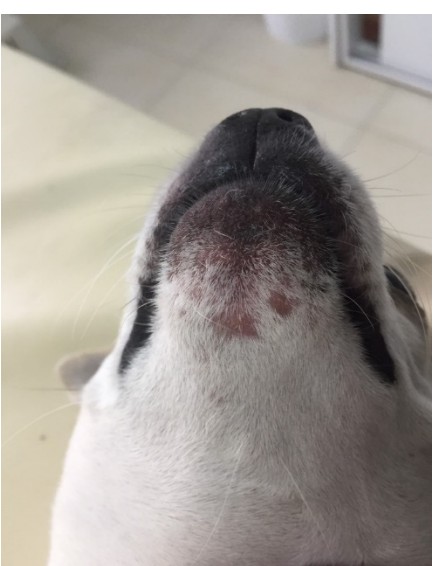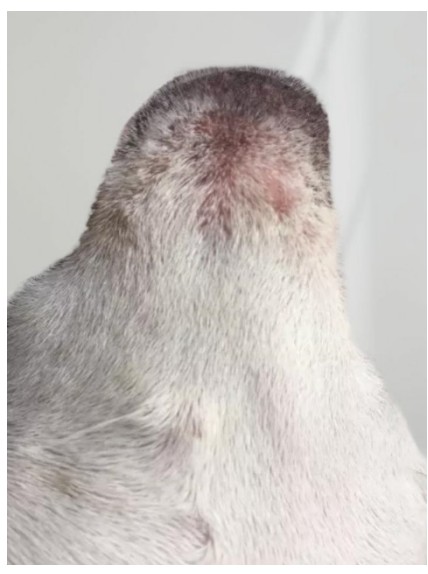

**Figure 4.** Treatment results of patient 4 with SILVER STOP® cream.

Any irritations caused by the cream were not reported in treated patients.

## 4. Discussion

The results of current studies correspond to those of other authors, who also demonstrate in vitro antimicrobial activity of preparations containing AgNPs against Gram-negative (*Escherichia coli*, *Pseudomonas aeruginosa*, *including multiresistant*) and Gram-positive (*Bacillus subtilis*) bacteria. Furthermore, when tested in a mouse skin infection model, bactericidal activity against pathogenic-resistant bacteria in in vivo infection was also proven. Products with AgNPs can be safely used as therapeutic agents in animal models [1]. It is found that silver nanoparticles bind to the surface of the bacterial cell, penetrate it, causing structural changes in the cell wall, disrupting its permeability, and the cell dies. In addition, they form free radicals in it, contributing to its death, also due to damage to the cell membrane, which makes it porous. Silver nanoparticles show effective antimicrobial properties due to their large surface area, which provides better contact with microorganisms [18]. An essential reason for the different sensitivity of Gram-positive and Gram-negative bacteria to AgNPs is the difference in the thickness and molecular composition of the membrane structures. The bactericidal activity is probably due to a change in the structure of the bacterial cell wall as a result of interactions with AgNPs, leading to increased membrane permeability and finally death [33]. AgNPs also react with sulfur- and phosphorus-rich biomolecules such as proteins, DNA, or membrane proteins that affect cell respiration, division, and ultimately its survival. Silver ions penetrate the cells, leading to aggregation of damaged DNA and thus affecting protein synthesis [18].

Our results are also in line with those of [34,35], which found moderate direct antibacterial properties against a spectrum of Gram-positive and Gram-negative bacteria, as well as improvement of immune functions at different levels in the body after administration of *P. sidoides.* Ethanol and acetone extracts of the roots of *P. sidoides* inhibit the growth of *Haemophilus influenzae*, *Moraxella catarrhalis*, and *Streptococcus pneumoniae* at a concentration of $5 \times 10^3$ mg·L$^{-1}$, as well as of *Aspergillus niger* and *Fusarium oxysporum* at a concentration of $5 \times 10^3$ mg·L$^{-1}$ [35]. The bioactive phytochemical constituents of *P. sidoides* may not exert a direct antimicrobial effect, but act by inhibiting microbial binding to host cell receptors, as well as by inhibiting key enzymes and the production of antimicrobial effector molecules such as nitric oxide and interferons by host cells [24].

The antimicrobial and antioxidant effects of *S. nigra* extract are probably due to its high phenolic content [36]. It can be used as an inhibitor of microbial growth. The antimicrobial properties of elderberry liquid extract against human pathogenic bacteria, and also against influenza viruses, are mainly due to phenolic compounds [37]. In addition, cysteine-rich peptides with activity against various Gram-negative bacteria, which exercise their effect by disrupting their bacterial membrane, are identified in the extract.

The results of the current research correspond to those of Süntar et al. [38], which prove that ethanol extracts of *H. perforatum* demonstrate strong antibacterial activity (values of MIC 8 µg·mL$^{-1}$) against *S. sobrinus* and *L. plantarum*, and exhibit moderate activity against *S. mutans* and *E. faecalis* at concentrations of 32 and 16 µg·mL$^{-1}$, respectively [39]. They establish that the water-soluble components are responsible for the antibacterial action of the plant, and support the recommended use of *H. perforatum* as a natural antibacterial agent in oral care products. Antimicrobial effect of *H. perforatum* extract against three clinical pathogens was demonstrated; *C. albicans*, *E. coli* and *S. aureus*, with the highest antimicrobial activity was found against a strain of *S. aureus* [39,40].

The results of our clinical trials (in vivo) on patients with various skin diseases show that it has a significant therapeutic potential, comparable to that of other medicinal ointments applied for the same purpose. After local treatment of the skin lesions for the required duration, healing was achieved in all patients using this cream alone, without any skin irritation after its application or other side effects. Our results show that through regular application of the cream, skin diseases of different nature and etiology can be successfully treated, such as dermatomycoses (microsporia, ringworm) and bacterial pyoderma superficialis, as well as even hepatocellular adenoma. Of course, further studies on a larger number of patients are needed to confirm additionally these results, as well as

testing the cream in a larger number of skin diseases. The results obtained in dogs show that the application of SS® cream in other animals and humans would be equally successful and promising. This herbal product is completely harmless and at the same time, has a significant antibacterial and antifungal effect in vitro and in vivo. It is a successful synergistic combination of silver nanoparticles and herbs with broad-spectrum antimicrobial action and tissue regenerating effect.

Our preliminary in vitro studies [6,7,17] have shown that the antimicrobial action of the product is mainly due to the herbal extracts it contains, as well as to the colloidal nanosilver in the final concentration applied (it is also determined as a result of preliminary such studies). The combination of these plant extracts and colloidal nanosilver prepared in this way is synergistic in terms of antibacterial and antifungal activity. Our present in vivo patient application results confirm the in vitro results obtained. In these clinical cases, no acute allergic and toxic reactions were observed after topical usage of SS® cream.

Apparently, it represents a successful synergistic combination of silver nanoparticles and herbs with broad-spectrum antimicrobial action and tissue regenerating effect. Based on these in vitro and in vivo results, we recommend its widespread application in practice.

## 5. Conclusions

The tested antimicrobial product SILVER STOP® cream (SS® cream) containing extracts of *P. sidoides*, black elderberry (*S. nigra*), and *H. perforatum* in colloidal nanosilver exhibited significant antimicrobial activity in vitro. When examined by the agar-gel diffusion method, the tested strains of *E. coli* showed the highest sensitivity, and the lowest—those of *S. aureus*. When in suspension with a density of $10^4$ cells·mL$^{-1}$, *E. coli* died quickly under the action of the cream—for 1 min. *S. aureus* was fully inactivated after 1 h of presence of SS® cream. *C. albicans* was completely inactivated after 10 min by the cream. When the tested strains were at a hundred times higher concentration ($10^6$ cells·mL$^{-1}$), they proved to be more resistant. In this concentration, the tested strains of *E. coli* died completely after 10 min under the action of SS® cream. Only single cells of *S. aureus* survived 2 h. The tested strains of *C. albicans* were completely inactivated after 1 h by SS® cream. All the results showed high antimicrobial activity of SS® cream.

**Author Contributions:** Conceptualization, T.P.P., T.E.P., M.D.K. and S.D.K.; methodology, M.D.K. and T.P.P.; validation, T.P.P. and I.I.; formal analysis, S.D.K. and F.H.; investigation, T.P.P., T.E.P. and M.D.K.; resources, T.P.P. and I.I.; writing—original draft preparation, I.I. and T.P.P.; writing—review and editing, T.E.P. and M.D.K.; visualization, T.P.P., T.E.P. and M.D.K. All authors have read and agreed to the published version of the manuscript.

**Funding:** This research received no external funding.

**Institutional Review Board Statement:** Not applicable.

**Informed Consent Statement:** Not applicable.

**Data Availability Statement:** Not applicable.

**Conflicts of Interest:** The authors declare no conflict of interest.

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
