# Peer review of "Antimicrobial Activity In Vitro of Cream from Plant Extracts and Nanosilver, and Clinical Research In Vivo on Veterinary Clinical Cases"

_cosmetics, doi:10.3390/cosmetics9060122_

Round 1

Reviewer 1 Report (Previous Reviewer 1)

This manuscript describes an in vitro and in vivo study of a SILVER STOP® cream containing herbal extracts in colloidal nanosilver (AgNPs), regarding its antibacterial properties as well as its therapeutic ability against various bacterial and fungal infections.

In my opinion, the revised manuscript improved as the authors took under consideration all my suggestions and now, I think that the manuscript is suitable for publication after minor corrections.

Minor comments:

The use of the term “Researched cream” is not suitable. I propose to replace the first sentence of the abstract as followed: “The antimicrobial effect of a cream containing extracts of African geranium (Pelargonium sidoides DC.), black elderberry (Sambucus nigra L.) and St. John's wort (Hypericum perforatum L.) in colloidal nanosilver (AgNPs) at a concentration of 30 ppm (denoted as SILVER STOP® cream) was examined in vitro.” Thus, the term “SILVER STOP® cream” or simply “cream” can be used throughout the manuscript.

Please remove some Russian words or characters in line 67.

Please present the units in homogeneous manner (cells.mL-1 or cells/mL). Also, in line 116 please correct the number of cells to “2 x 106”.

Author Response

This is the file for the first reviewer.

Reviewer 2 Report (New Reviewer)

The authors presented their research finding titled “Antimicrobial Activity in vitro of Cream from Plant Extracts 2 and Nanosilver, and Clinical Research in vivo on Veterinary Clinical Cases” is well written and they come out with an effective anti microbial cream for the animal usage. However, the reviewer suggest few corrections before it is accepted in cosmetics.

Minor Commnets:

  1. The article needs careful grammar and spelling check.

Major Comments:

  1. The author used the plants crude extracts for the preparation of cream. It would be beneficial if it is mention about solvent used for the extraction. 
  2. What is the yield of crude extract in each plants sample. What is the initial raw materials taken for extraction..etc.,
  3. Why only the 30 ppm concentration used in this study?
  4. The anti-microbial product method is not clear? The reviewer suggest to revise the method with clarity regarding the plant extract proportion or concentration ratio.
  5. In animal treatment model, each animal defect are carried out at different disease conditions. it would be beneficial if the author discuss the pathogen responsible for such disease in the discussion.

Note: Looking at the figure 4. The animal handling personal didn’t followed the hygiene protocol while using the animals (The nail grown hand). The reviewer strongly suggest to follow the animal ethics protocol while handling the animals.

Author Response

This is the file for the second reviewer.

This manuscript is a resubmission of an earlier submission. The following is a list of the peer review reports and author responses from that submission.

Round 1

Reviewer 1 Report

This manuscript describes an in vitro and in vivo study of a SILVER STOP® cream containing herbal extracts in colloidal nanosilver (AgNPs), regarding its antibacterial properties as well as its therapeutic ability against various bacterial and fungal infections.

In my opinion, the experimental results seem sound and, in general, this work could be of interest for researchers working on the field. However, this manuscript lacks of novelty since there are many works in the literature describing analogous studies. Therefore, the authors should better present the novelty of the study focusing on the importance of their findings. Based on this I suggest to the authors to explain in more details their results. Specifically:

1.      Include pictures of plates with bacteria treated with the cream in order to prove the measured zones of inhibition.

2.      Please specify the incubation time for each bacteria strain.

3.      Please the dates of dog treatment. For example in Figure 1 the initial photos were taken on 30-5-2022 and next on 20-5-2022.

Additionally, the discussion section, the benefit of this product should be presented in comparison with other similar products.

Also some details regarding the preparation of SILVER STOP® cream should be added in the experimental section.

Author Response

In attached file is the response for the reviewer 1

Reviewer 2 Report

The paper presented for review takes up a very interesting topic. It is especially relevant in the face of the growing resistance of bacteria to the used antibacterial agents and antibiotics.

However, the manuscript requires a number of corrections. And the most important of these is a clear indication of the limitations of this study. At the moment, especially with the extremely limited discussion, the work does not seem to be objective.

Abstract:

It requires corrections: remove the name of the colloidal silver manufacturer (such details are given in the methodology section) and describe exactly what research was carried out with the most important results with numerical values and statistical significance.

Key words:

Delete the word: SILVER STOP® cream; is a trade name - not present in the MeSH database, do not use the terms that appear in the title of the paper as keywords, remove the word: dogs - look for an appropriate term in the MeSH database in the field of veterinary therapy.

Introduction:

“The dose of colloidal nanosilver for oral use is 5 μg per 1 54 kg body weight [14].” - incorrect source. Check again throughout the manuscript whether the cited sources relate to research papers that address the topic. Currently, both in the introduction and in the discussion, similar data are shown. Consider organizing this information

Aim of the study:

No information about fungi tests. The goal of casuistic veterinary studies was not indicated either

Materials and methods:

Antimicrobial drug. - is the tested cream a drug or a cosmetic? What is the legal status of this product?

Statistical analysis;

Describe exactly how the statistics were compiled.

Line 174: Why was a different font used here? Patent information has been included before, please delete this sentence.

Table 1 and table 2: unnecessarily large

Table 3 - requires editing correction (see verse 3), unnecessarily large

Line 213: Make some editing changes to make the text easier to read.

Photo-based results: improve aesthetics, reduce photo size and highlight diseased areas

The first sentence of the discussion refers to the research, but these items have not been mentioned in the literature.

Writing: in vivo - use italics

Notation: Our results are also in line with those of [34], - is invalid

Notation: The results of the current research correspond to those of Süntar et al. (2016), - is invalid

Conclusions: This chapter needs attention. The last paragraph is an overinterpretation.

Additionally, the work requires one more chapter: study limitation. Please indicate what limitations the authors of this paper see. What limitations in the interpretation of these results must be kept by the reader, especially in the transposition of in vitro results into practice, and case studies for the overall picture. Please note that the cream is not only active ingredients but also a whole range of ingredients that build the base of the cream. Many of them, especially surfactants, can have antimicrobial effects.

There is no paragraph on conflict of interest and research funding sources

Author Response

In the attached file is the answer for the Reviewer 2

Round 2

Reviewer 1 Report

The authors took under consideration almost all my suggestion and now I think that the manuscript is suitable for publication. Only one comment the new Figure 1 is not suitable for publication. It would be better to be deleted.

Author Response

Thank you. We deleted this figure from the manuscript.

Reviewer 2 Report

Title

Remove the proprietary name from the title and emphasize the use of silver nanoparticles here.

SILVER STOP is used 49 times in the manuscript. Us an abbreviation e.g.: Study cream (SC)

Abstract

Sentence: The diameters of the inhibitory zones were between 15.3 + 0.4 and 19.0 + 2.0 mm. – remove. Give information about statistical results.

Sentence: It can be used successfully as a harmless herbal medicine for external treatment of skin diseases including bacte rial and fungal infections. Remove. This sentence is an over-interpretation

Background

Line 48: to improve the activity of dry plants - Dried plants are used very rarely as pharmaceutical or cosmetic raw materials. Various types of plant extracts are used more often. Therefore, nanotechnology does not improve the activity of dried plants but the active ingredients obtained from them.

Line 58: The dose of colloidal nanosilver for oral use is 5 μg per 1 58 kg body weight [14]. The second time I indicate that the given refference is incorrect. Show data on toxicological studies and studies that estimate of safe doses in humans. If such estimation was not made, indicate the dose considered as safe.

For skin conditions conditions, the colloidal nanosilver dosage is 0.01 mg.m-2 [16]. What does it mean: skin conditions?

The dose to be used for SILVER STOP® 61 cream is 0.0092 mg.m-2 [17]. Explain that the dose indicated is the amount of Strebra in a one-time application.

Line 66: A very good track record is achieved by African Geranium, Black Elderberry, St. John's Wort, etc. [17,19-21]. Indicate the correct systematic names, including the Latin three-part name, for each plant mentioned when you talk about it for the first time. Characterize these plants in the same order you listed them. African Geranium, Black Elderberry: check whether these names should be capitalized in English.

Line 86: Further studies are 86 needed in this direction, especially on patients with skin infections, where the applica-87 tion of herbal remedies seems particularly promising.: remove

Line 90: remove: SILVER STOP®. Pelargonium sidoides DC. - only the first time a plant is mentioned, the tri-element notation is used. Use the appropriate shortcuts. For this plant: P. sidoides.

Remove the sentence: We also set out to conduct an initial clinical trial of the cream for its healing effect in patients with common skin infections!. describe that the second goal of the study was to indicate the usefulness of the preparation through the description of veterinary cases.

Materials and Methods

Line 98-101: correct the editing and recording form of systematic names shown again

Line 106: when writing about an organism (bacteria) for the first time, use its full systematic name

Line 160 – use some spaces between lines to make the text more readable

Statistical analysis: this chapter should be before descriptions of veterinary patients! Start by showing that the results are shown as means and standard deviations. Write with what program the statistics were counted. In the methodological part it was written that the experiment was repeated 3 times. Was the student's t-test counted for 3 results in each group?

Results

Figure 1 is redundant. Remove it. If it contains an important element that was not mentioned in the description of the results - explain it.

There is no information about the results of the statistical analysis in the results section. I understand that such an analysis was not performed, but only the diameters of the inhibition zones for the tested cream and the reference drug were compared. If this is the case: describe it in the chapter: material and method and remove the information about the statistical analysis, since it was not performed.

Figure 2. In my opinion, the photos add nothing to this manuscript. Move them to supplementary materials. Figure 2 should be removed from the manuscript.

Discussion

Lines 288-298 – this is Background not a discussion of the results.

Line 334 – in vivo – use italics

Line 349: In conclusion: those results suggests that herbs or silver are the main active ingredient?

Line 353 - Our pre-353 sent in vivo patient application results confirm the in vitro results obtained as well as the safety of the product. Remove! The safety of the product was not studied. It can only be indicated that in these clinical case descriptions no acute allergic and toxic reactions were indicated.

Line 355: We are convinced that the wide application of the product in practice would be very successful and would confirm these results. Remove!. Science is not based on inner beliefs but on evidence

Conclusions

Remove the word: drugs from the conclusion. Delete the last sentence as overinterpretations. Do not use your own name but the description of the active ingredients of the tested cream.

Conflict of interests

in this chapter it should be indicated that some authors are employees of the manufacturer of the tested cream.

Study limitation

As I asked in the first review - add a study limitation chapter. Describe what could have influenced the obtained results. What is the importance of a cosmetic base here? Whether its ingredients (especially preservatives and emulsifiers) could influence the growth of microorganisms?

Author Response

Thank you for your valuable advice. Please find it in the attachment.

Round 3

Reviewer 2 Report

Unfortunately, despite the information contained in the responses that the authors corrected or tried to correct the manuscripts, the presented manuscript does not contain the corrections I wrote about earlier.

The name of the cream as a proper name should appear only in the Materials and Methods chapter. It appears three times in the abstract itself.

Pictures of the described veterinary cases: if the authors insist that they are included in the manuscript, they should be described in detail in the chapter: Results.

There is still no chapter: study limitations (when it is indicated in the responses that such an amendment had been applied - or rather, that the authors tried to apply it). Once again, please add a chapter: study limitations.

Black Elderberry - please write in lowercase, similarly the word "wort" - please write in lowercase.

Two abbreviations have been introduced: SSC and RC - their meaning is unclear.

There is no need to enter the software code, I asked for the manufacturer and country of production. These data are still missing.

The results include data on statistical analysis. Please let me know whether the indicated statistical test can be used for such a small amount of data (3 repetitions).

Please complete table 1 with data from statistical analysis. Please provide the exact p-values, not just an indication of whether they are less than or greater than a predetermined limit.

Tables 2 and 3 lack the full names of the microorganisms. Provide them in the table itself or in the additional information (title or information below the table).

There is no discussion of the described veterinary cases. After all, strebro has been tested many times for this application. Why do the authors not cite such studies?

The summary has not been corrected enough.

I disagree with the conflict of interest information. Employment at the manufacturer of the tested preparation is a conflict and it should be proved. Information should also be given to what extent the manufacturer was involved in the design, analysis and elaboration of the results of this test.

Author Response

The authors submit the final edition
